# Antigen Load and T Cell Function: A Challenging Interaction in HBV Infection

**DOI:** 10.3390/biomedicines10061224

**Published:** 2022-05-24

**Authors:** Ilaria Montali, Andrea Vecchi, Marzia Rossi, Camilla Tiezzi, Amalia Penna, Valentina Reverberi, Diletta Laccabue, Gabriele Missale, Carolina Boni, Paola Fisicaro

**Affiliations:** 1Laboratory of Viral Immunopathology, Unit of Infectious Diseases and Hepatology, Azienda Ospedaliero-Universitaria di Parma, 43126 Parma, Italy; ilaria.montali@unipr.it (I.M.); avecchi2@ao.pr.it (A.V.); marzia.rossi@unipr.it (M.R.); camilla.tiezzi@unipr.it (C.T.); apenna@ao.pr.it (A.P.); vreverberi@ao.pr.it (V.R.); diletta.laccabue@unipr.it (D.L.); gabriele.missale@unipr.it (G.M.); 2Department of Medicine and Surgery, University of Parma, 43126 Parma, Italy

**Keywords:** chronic HBV infection, T cells, antigen load, HBsAg

## Abstract

Current treatment for chronic HBV infection is mainly based on nucleos(t)ide analogues, that in most cases need to be administered for a patient’s lifetime. There is therefore a pressing need to develop new therapeutic strategies to shorten antiviral treatments. A severe dysfunction of virus-specific T cell responses contributes to virus persistence; hence, immune-modulation to reconstitute an efficient host antiviral response is considered a potential approach for HBV cure. In this perspective, a detailed understanding of the different causes of T cell exhaustion is essential for the design of successful functional T cell correction strategies. Among many different mechanisms which are widely believed to play a role in T cell dysfunction, persistent T cell exposure to high antigen burden, in particular HBsAg, is expected to influence T cell differentiation and function. Definitive evidence of the possibility to improve anti-viral T cell functions by antigen decline is, however, still lacking. This review aims at recapitulating what we have learned so far on the complex T cell–viral antigen interplay in chronic HBV infection.

## 1. Introduction

CD8 T cells are pivotal elements of the antiviral immune response. Following recognition of antigen exposed on antigen-presenting cells (APC), CD8 T cell activation and differentiation can result either into a fully functional response with T cell memory generation, which is key to controlling infection, or to dysfunctional T cell responses of different severity, up to exhaustion and even physical T cell deletion if the pathogen is not successfully eliminated and antigen exposure persists overtime. T cell exhaustion during persistent viral infections is a well-documented phenomenon, which is characterized by the progressive and hierarchical loss of antiviral functions [1]. Among many different factors that have been described to play a role in T cell exhaustion, persistently high antigen expression levels can represent either the cause or the consequence of T cell dysfunction.

Hepatitis B virus (HBV) infection targets the liver, which represents a unique immunological environment enriched with different antigen presenting cell populations which contribute to its tolerogenic properties [2]. Thus, the fate of CD8 T cells primed within the intrahepatic environment as well as the T cell response to the elevated antigenic burden constantly present during chronic HBV infection (CHB) have been the subject of various experimental analyses both in animal models and in patients. Additional studies are needed, however, to address a series of still unsolved issues which make the topic a matter of debate within the scientific community.

As many other viruses, HBV exploits the production of a large excess of viral antigens as a mechanism to evade the host immune response [3]. Among HBV proteins, Hepatitis B surface antigen (HBsAg) is present in the form of noninfectious subviral particles, in a large excess with respect to infectious virions [4], and largely outnumbers the other viral proteins. Persistent exposure to high HBsAg titers is believed to represent a cause of immunological impairment in chronic HBV infection. Instead, sustained HBsAg clearance from the blood, with or without detection of anti-HBs antibodies, is a rare event in chronic HBV infection and an accepted marker of infection resolution, currently referred to as “functional cure” [5].

Understanding the effect of HBsAg over-expression on host immune responses can provide important information in the perspective of devising new therapeutic strategies for chronic hepatitis B and novel reliable biomarkers to monitor the effect of therapy and the evolution of infection. In this perspective, this manuscript will analyze studies aimed at improving our knowledge of the causal link between HBsAg expression and T cell dysfunction in chronic HBV infection, with the awareness that this represents just one among numerous factors playing a role in T cell exhaustion.

## 2. HBV Proteins

HBV virions are composed of an icosahedral nucleocapsid, formed by core proteins (HBcAg) containing partially double-stranded DNA (rcDNA: relaxed circular DNA) and a viral polymerase. The nucleocapsid is surrounded by the envelope, consisting of the surface antigen (HBsAg). Six polyadenylated transcripts derive from the viral genome, including the pregenomic RNA (pgRNA), and seven proteins [4]: HBeAg (secreted e antigen), HBcAg, HBV Pol/RT (polymerase, with reverse transcriptase activity), PreS1, PreS2, HBsAg, and HBx (antigen x, transcription regulator, necessary for the initiation of infection [6]).

The HBV envelope consists of three proteins: the large (L), middle (M) and small (S) proteins, originated by starting translation at different sites of the HBV genome. The small (or major) protein contains only the HBs antigen, which is referred to as the S domain; the middle protein contains an additional 55 AA sequence that represents the PreS2 antigen and finally the large protein, in addition to HBs and PreS2, is composed of an additional antigen sequence which is called PreS1 [4]. Besides composing the infectious virions, these surface proteins also assemble to form non-infectious sub-viral particles (SVP) with a spherical or filamentous morphology, that are present in a large excess over the complete virions. All viral antigens derive from the transcription of the covalently closed circular HBV-DNA (cccDNA), which is present in the nucleus of the infected cell as a mini-chromosome. In addition, part of the HBV genome can be found in an integrated form in the host’s genome, especially in HBeAg-negative patients, giving rise to HBsAg and HBx transcription, but not to infectious virions [7,8,9,10,11] (Figure 1). Current immunoassays employed to detect or quantify conserved HBsAg epitopes within the S domain do not distinguish between the different forms or origin of envelope proteins. HBsAg detection is routinely used for the diagnosis of HBV infection and HBsAg seroclearance, along with anti-HBs antibodies appearance, indicates immunological control over the infection. Conversely, persistent serum HBsAg detection is associated with chronic HBV infection, with HBsAg levels indirectly reflecting the expression activity of viral cccDNA or of integrated HBV-DNA fragments. Thus, HBsAg seroclearance represents the desired outcome of current antiviral treatments. Hence, serum HBsAg titration is commonly used to monitor patients undergoing nucleos(t)ide analog (NUC) therapy and to predict the response to PegIFN treatment [12].

The precore/core gene codes for an amino acid sequence corresponding to the whole core protein plus 29 residues. The N-terminal 19 amino acids represent a signal peptide that drives the newly synthesized protein toward the secretory pathway. Following removal of the signal peptide and further proteolytic cleavages at the C-terminal end of the precore sequence, a 17 kDa protein is generated which is the predominant secreted moiety of the pre-core gene translation that is called e antigen [13,14]. Interestingly, an immunoregulatory role with tolerogenic properties has been proposed for this antigen [15,16]. Finally, core and polymerase proteins derive from the translation of the pre-genomic(pg)RNA.

Recently, the simultaneous detection of serum HBcAg, HBeAg and p22 precore-derived proteins, all sharing an identical 149 amino acid sequence, together referred to as Hepatitis B core-related antigen (HBcrAg), has been proposed [17] (Figure 1) as a potential surrogate marker of cccDNA detection, since elevated serum HBcrAg levels have been found to correlate with the transcriptional activity of the cccDNA in the liver, highlighting the presence of more active viral replication in HBcrAg-positive patients [18,19]. In line with this conclusion, several studies have described a better correlation of HBcrAg with cccDNA than quantitative HBsAg, and the decline of HBcrAg has been reported to reflect the behaviour of cccDNA in patients receiving nucleos(t)ide analog therapy [19,20,21,22]. A limitation of the available HBcrAg test is its low sensitivity, with a lower limit of quantification of about 3 log_10_ U/mL, that would preclude its application to patients with low viral replication. However, new and more sensitive tests are being developed [23], which can make HBcrAg detection applicable to a wider range of clinical conditions.

## 3. The Role of Antigen Expression in Animal Models

Animal models of infection have been used to elucidate the interdependence of antigen load and T cell exhaustion in persistent infections.

The amount of antigen and the duration of antigen presentation were shown to negatively influence the T cell function, ultimately leading to impaired viral clearance in mice with chronic LCMV-CL13 infection [24]. In this model, chimeric mice lacking MHC class I expression on non-hematopoietic (parenchymal) cells displayed viral loads 10-fold higher than wild type (wt) mice as a result of a lack of infected cell clearance by CTL. This could lead in 4–6 weeks to a decline in T cell numbers and function, despite initially greater CD8 T cell cytokine production. In this model of LCMV chronically infected mice, more severe degrees of T cell dysfunction were associated with higher levels of antigen. Moreover, the level of T cell exhaustion to individual epitopes was proportional to the amount of peptide available for presentation to CD8 T cells, up to physical T cell deletion induced by epitopes persistently presented at higher levels [25]. A progressive decline of CD8 T cell responses, up to their attrition, was also reported in LCMV-infected mice that were unable to clear the infection because of CD4 T cell depletion, compared to mice with a finite duration of infection, either acute or protracted [26]. These results were supported by studies in other viral models, such as influenza A virus infection in mice, where repeated antigen stimulation, twice a week for at least 4 weeks, caused reduced frequency and function of CD8 T cells compared to single-primed control mice [27].

These studies, however, did not specifically address the main immunological issue related to hepatotropic infections, i.e., the antigen presentation to T and B cells within the peculiar tolerogenic environment represented by the infected liver [2]. To elucidate the role of the intrahepatic antigen load, a recombinant adeno-associated viral vector system (rAAV8 transduction) was used to induce antigen expression within the liver of B6 mice. Administration of a high dose of viral vector, leading to antigen expression in 100% of the hepatocytes, induced CD8 T cell silencing between week 1 and 3, along with PD1 and Tim3 up-regulation, despite the initial development of antiviral effector functions. Conversely, when viral antigens were expressed in a percentage of hepatocytes lower than 25%, as a result of a lower dose of viral vector administration, long term CD8 T cell function was maintained, identifying in this way a threshold level of antigen load within the liver for optimal and efficient T cell induction and differentiation [28]. Similarly, adoptively transferred naïve HBV-specific T cells in HBV transgenic mice with viral antigen expression in all hepatocytes were unable to develop effector functions and this inhibition was mediated by PD1 signaling [29].

The notion that high levels of antigen expression can induce T cell dysfunction in CHB has recently been challenged, taking advantage of different mouse models, allowing the expression of variable levels of antigen within the liver, and the activation of different pathways of antigen presentation by hepatocytes, or by Kupffer cells (KC) and hepatic dendritic cells upon transduction with a LCMV-based vector. As expected, HBV-core expression by 100% hepatocytes led to CD8 T cell dysfunction, but the scenario did not change when the amount of hepatocellular antigen was dramatically reduced by more than 15-fold. Instead, antigen-transfected KC induced a correct T cell differentiation into effector cells, with a completely divergent transcriptional and chromatin-accessibility profile. Thus, hepatocyte T cell priming rather than high antigen levels *per se* would promote a defective differentiation program that became partially irreversible and gradually triggered an exhaustion profile [4,30]. These conclusions were further supported in a HBV-replication-competent transgenic mouse model where number and phenotype of antigen-specific T cells isolated from the liver of animals with elevated antigen titers were similar to those of animals that succeeded in clearing HBsAg over time and reached undetectable antigen levels [31].

In line with these findings, in two different immune-competent mouse models of persistent HBV infection expressing clinically relevant HBsAg titers (10^3^–10^4^ IU/mL), 1–2 log_10_ antigenemia decline induced by small interfering RNA compounds targeting all HBV transcripts (shHBV) did not result in improved CD8 T cell function. However, decline of HBV antigen expression by RNA interference, but not inhibition of HBV replication by NUC treatment, allowed a significantly stronger T cell response with control of infection upon a subsequent heterologous therapeutic vaccination (TherVacB) with protein priming followed by Modified Vaccinia Ankara (MVA) virus vector boosting [32]. Since TherVacB alone could not induce the same results, these observations support the notion that preliminary antigen load reduction can be extremely helpful for a successful therapeutic vaccination, as also shown in a previous report in AAV/HBV carrier mice, where reconstitution of the B and T cell responses was reached through an HBsAg vaccination, composed of the CpG-adjuvanted Engerix B preventive vaccine (EnxB/CpG), preceded by HBsAg sero-clearance by HBsAg-neutralizing antibodies administration, but not by either approach alone [33]. This contrasts with observations in SHIV infected macaques, where potent and broadly neutralizing anti-HIV1 antibody (bNAb) administration, but not anti-retroviral therapy (cART), led to CD8 T cell-mediated control of viremia in 6 out of 13 animals [34]. Of note, early bNAb administration at the start of acute infection increased the likelihood of success, suggesting the importance of infection duration in the induction of T cell exhaustion.

## 4. Human Studies

An improved T cell reactivity can be detected during NUC therapy in both HBeAg+ and HBeAg− CHB patients [35,36,37,38], and a hierarchy of T cell functional responsiveness has been observed among infections with different degrees of virus control, with better T cell functionality in patients who succeed in clearing serum HBsAg and worse antiviral T cell responses at the opposite extreme of the HBV-related disease spectrum, namely in viremic untreated chronic patients [36]. Envelope-specific T cell responses can be more easily detected only after HBsAg elimination, but a clear correlation between serum HBsAg titers and in vitro T cell function in chronic infection has not been univocally reported, although the common view is that persistently elevated HBsAg levels play a significant role in the pathogenesis of T cell dysfunction. The difficult detection of HBsAg-specific T cells in chronic HBV patients as well as the difficult recovery of the envelope-specific T cell function upon checkpoint inhibition support this possibility [36,39,40,41].

In particular, a recent study performed in a wide population of both HBeAg+ and HBeAg- CHB patients, broadly distributed in terms of age and including also very young patients, suggested that the duration of HBsAg exposure, rather than antigen concentration *per se*, represents a key determinant of HBs-specific T cell dysfunction, inducing progressive decline of HBs-specific T cells as the infection persists over time, up to their disappearance in older patients. Since the vast majority of CHB patients got infected in childhood, robust HBV-specific T cell responses can more frequently be detected within the first three decades of life, with a direct correlation between HBs-specific T cell numbers and serum HBsAg levels. In this study, a multivariate linear regression analysis evidenced patients age as the only factor significantly associated with HBs-specific T cell frequencies, underlying the importance of infection duration influence on the T cell function. Thus, the combination of high antigen loads with long-term exposure represents the detrimental factor for antigen-specific T cells, but not for total T or NK cell populations, mitigating the possibility that HBsAg itself exerts a direct immunosuppressive role [41].

Partially at variance with these results, a threshold of HBsAg level <500 IU/mL has been proposed to associate with greater polyfunctional CD4 T cell responses and to predict a positive effect of PD1/PDL1 blockade on HBV-core, but not HBV-envelope specific T cells [42].

In the search for serological biomarkers of T cell functional efficiency, serum HBcrAg has recently been reported to represent a more reliable predictor of response to checkpoint inhibition than HBsAg quantification and to correlate better than HBsAg with the strength of the T cell response to HBV core and polymerase [43]. Since HBcrAg has been shown to reflect the epigenetically active state of the viral cccDNA [22], it is not surprising that detection of HBV core- and polymerase-specific T cell responses are associated with lower HBcrAg levels, which indicate reduced HBV antigen transcription and burden. Instead, widespread expression of integrated HBV-DNA can maintain high HBsAg levels and HBsAg-specific T cell exhaustion even when cccDNA activity is controlled [8,9,10].

Length of infection, patients’ age, HBsAg levels, NUC therapy starting point and duration have been proposed as components of an algorithm to predict the likelihood of HBsAg decrease or loss and of HBV-specific T cell functional restoration upon NUC discontinuation [44].

## 5. Conclusions

Production of a large excess of antigens by HBV is widely believed to represent a strategy to deceive the host immune defenses by diverting the neutralizing antibody activity away from virions [3] and by progressively tolerizing HBV-specific T cells. In addition, a direct suppressive interaction between HBsAg and innate immunity components, such as dendritic cells and monocyte/macrophages, has been reported to impair the T cell stimulatory activity of these APCs [45,46]. Initial evidence of an improved T cell reactivity in virus-suppressed NUC-treated chronic patients, especially after HBsAg seroclearance, led to the hypothesis that HBsAg could exert direct inhibitory activity on the immune responses. The setting of NUC treatment, however, does not represent a good model to study the effect of antigen on T cell responses given the very slow and modest decline of antigen over time in most of the treated patients and the possible contribution of different additional factors to T cell functional reconstitution, in particular resolution of liver inflammation induced by therapy. With respect to this issue, more recent studies have provided a series of important observations. First, in patients with chronic HBV infection stratified by HBsAg levels, the overall immune cell populations, such as total T and B cells, do not differ significantly in phenotype and function according to the antigen load [41,42,43]. Instead, T and B cell functional inhibition is selectively antigen-specific [47,48,49], with a greater impairment of T cells specific for HBsAg, which is the most abundant antigen detectable in chronically HBV infected hosts, although also T cells specific for other HBV proteins display variable degrees of exhaustion. A second important point is that antigen-specific T cell impairment is not only a function of the antigenic load, but also of the duration of HBV infection and antigen exposure, with a clear correlation with patients’ age. Finally, HBcrAg has been reported to be more reliably correlated than HBsAg not only with transcriptional cccDNA activity but also with antiviral T cell responses, thus suggesting a combined use of these parameters, including levels of anti-HBs antibodies, for a more accurate CHB cure monitoring [50]. However, low HBsAg levels were demonstrated to retain a prognostic value, since they are significantly associated with a higher functional cure rate in case of PEG-IFNα treatment [12] and a higher likelihood of HBsAg seroconversion upon NUC suspension [51,52,53].

Although the effect of antigen decline on T cell function still remains a debated issue, strategies to lower the antigen burden before immune cell stimulation may be successful in optimizing lymphocyte responsiveness to exogenously administered antigens, for example by vaccine therapies. These strategies include RNA interfering compounds (RNAi) targeting viral transcripts [54], nucleic acid polymers (NAP) that inhibit HBsAg release [55,56,57], or anti-HBV humanized monoclonal antibodies [58] (Table 1).

Definition of extent and duration of viral protein decline sufficient to allow the restoration of a functionally efficient immune response will be also very important, as well as the identification of reliable biomarkers to predict possible virus rebounds. Finally, a reasoned combination of more than one approach to target different mechanisms of immune dysfunction and tailoring therapy also by patient age will have to be considered in the objective of extending functional cure to a larger proportion of infected patients. In particular, the possibility of an earlier start of therapy will have to be considered in the light of the concept that younger patients with shorter duration of exposure to the virus could likely get the maximal benefit from a successful immune rejuvenation.

## Figures and Tables

**Figure 1 biomedicines-10-01224-f001:**
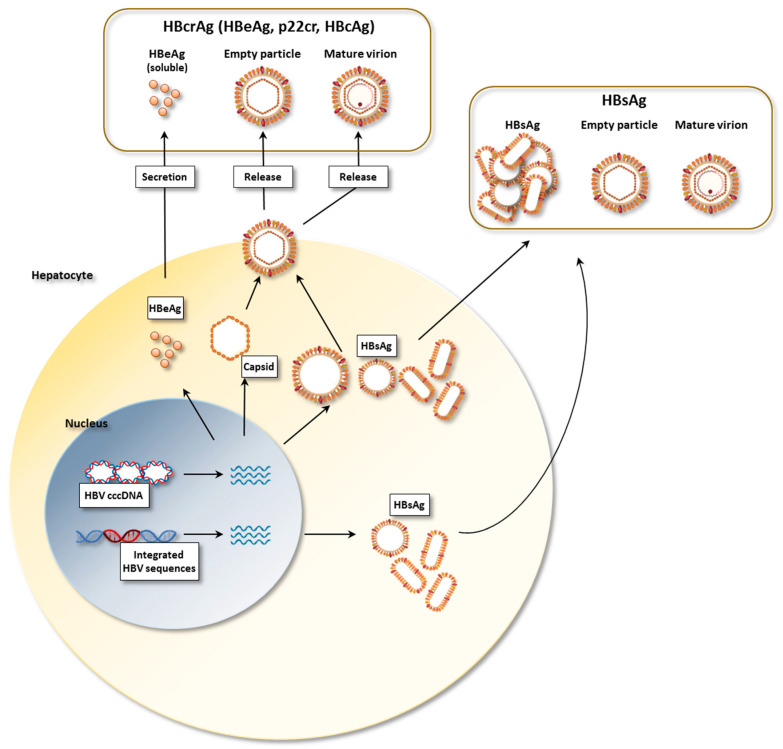
Schematic representation of HBsAg and HBcrAg production from the HBV-infected hepatocyte. Created with: BioRender.com (accessed on 20 May 2022).

**Table 1 biomedicines-10-01224-t001:** Novel approaches tested in clinical trials for the reduction of antigen load in chronic HBV infection.

Class of Agents	Drug	References
**Small interfering RNAs** **(siRNA)**	**ARC520**	[8,59,60]
	**JNJ-3989**	[61]
	**AB-729**	[62]
	**VIR-2218**	[63]
	**RG6346**	[64]
	**ARB-1467**	[65]
	**ARB-1740**	[66]
**Antisense Oligonucleotides** **(ASO)**	**BEPIROVERSEN**	[67]
	**GSK3389404**	[68]
	**RO7062931**	[69]
**Nucleic Acid Polymers** **(NAP)**	**REP 2139**	[70,71,72,73]
	**REP 2165**	[70]
	**REP 2055**	[73]
	**ALG 010133**	[74]
**Monoclonal Antibodies** **(Mab)**	**GC1102**	[75]
	**VIR-3434**	[76,77]

## Data Availability

Not applicable.

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
