# Peer review of "Antigen Load and T Cell Function: A Challenging Interaction in HBV Infection"

_biomedicines, 2022, doi:10.3390/biomedicines10061224_

Round 1
Reviewer 1 Report
General comment
This relatively short review gives a very good overview on the current understanding of chronic HBV infection, its monitoring by laboratory diagnosis and the most recent promising therapeutic approaches. Some minor points should be corrected.
Specific points
- Spell out here HBV
- L2. Explain HBsAg at the first mention.
- L26. Typo: sites, not sited
- L39-43. I suggest to reword these sentences and to add here some explanations which come only later. Disappearance of HBsAg and appearance of anti-HBs after acute HBV infection indicates immunological control over the HBV infection. Persistent infection leads to persistent HBsAg in serum, the level of which reflects indirectly the number and expression activity of HBV genomes or integrated HBV DNA fragments in the host. HBsAg seroclearance requires both decrease of HBV genomes and activation of the anti-HBs response.
- L10. Briefly explain what EnB/CpG is. Is this similar to Heplisav-B?
- L30-41. The main point of impaired T cells immunity in CHB is the time point of infection, i.e., perinatal or within the first years of life. Of course, this leads also seemingly to the role of the duration of the infection. The discussion of the role of duration of the HBV infection is partly misleading or irrelevant.
- Table 1 would appear nicer without the abbreviations of siRNA, ASO etc.
Author Response
We thank Reviewer 1 for her/his encouraging evaluation of our review and for very useful indications.
In the attached revised manuscript we spelled out HBV when first mentioned, as well as HBsAg and corrected the typo. We introduced some explanations about HBsAg detection meaning in acute and chronic HBV infection and also about the EnxB/CpG vaccine (which is the CpG-adjuvanted Engerix B preventive vaccine). Moreover, we rephrased the reference to the study by Bertoletti's group about the influence of antigen exposure and infection duration on the T cell function. We hope that this point is now clearer. Finally, also Table 1 has been corrected as indicated.
We believe that Reviewer's 1 suggestions helped a lot to improve the quality of our manuscript and hope that it is now acceptable for publication.
Best regards
Paola Fisicaro and Carolina Boni

Reviewer 2 Report
The authors reviewed the studies on the causal link between HBsAg expression and T cell 13 dysfunction in chronic HBV infection. It is interesting and helpful. I have no concerns with this review.
This review covered 78 manuscripts on HBsAg expression and T cell dysfun ction in chronic HBV infection. I can’t find any similar review on this topic. The review has been concisely written and easy to read. The read ers will be benefitted from this review to understand HBsAg expression a nd T cell dysfunction in chronic HBV infection.
Author Response
We thank Reviewer 2 for her/his very kind appreciation of our manuscript and attach the revised version for his/her knowledge.
Best regards
Paola Fisicaro and Carolina Boni
